# Gait Stability Measurement by Using Average Entropy

**DOI:** 10.3390/e23040412

**Published:** 2021-03-31

**Authors:** Han-Ping Huang, Chang Francis Hsu, Yi-Chih Mao, Long Hsu, Sien Chi

**Affiliations:** 1Department of Electrophysics, National Yang Ming Chiao Tung University, Hsinchu 30010, Taiwan; samuel_hphuang@nctu.edu.tw (H.-P.H.); francis-920@hotmail.com (C.F.H.); long@cc.nctu.edu.tw (L.H.); 2Center for Industry-Academia Collaboration, National Yang Ming Chiao Tung University, Hsinchu 30010, Taiwan; ycmao@nycu.edu.tw; 3Department of Photonics, National Yang Ming Chiao Tung University, Hsinchu 30010, Taiwan

**Keywords:** complexity, disorder, entropy of entropy, average entropy, gait analysis, gait stability

## Abstract

Gait stability has been measured by using many entropy-based methods. However, the relation between the entropy values and gait stability is worth further investigation. A research reported that average entropy (AE), a measure of disorder, could measure the static standing postural stability better than multiscale entropy and entropy of entropy (EoE), two measures of complexity. This study tested the validity of AE in gait stability measurement from the viewpoint of the disorder. For comparison, another five disorders, the EoE, and two traditional metrics methods were, respectively, used to measure the degrees of disorder and complexity of 10 step interval (SPI) and 79 stride interval (SI) time series, individually. As a result, every one of the 10 participants exhibited a relatively high AE value of the SPI when walking with eyes closed and a relatively low AE value when walking with eyes open. Most of the AE values of the SI of the 53 diseased subjects were greater than those of the 26 healthy subjects. A maximal overall accuracy of AE in differentiating the healthy from the diseased was 91.1%. Similar features also exists on those 5 disorder measurements but do not exist on the EoE values. Nevertheless, the EoE versus AE plot of the SI also exhibits an inverted U relation, consistent with the hypothesis for physiologic signals.

## 1. Introduction

Maintaining walking stability requires stable gaits. The cost of falls due to unstable gait is severe, especially for the elderly [1]. Therefore, gait stability measurement is critical.

In gait stability measurement, stride interval (SI) and step interval (SPI) time series are important parameters that are often used. Specifically, the SI is defined as the time from a heel strike of one foot to the next heel strike of the same foot. Similar to SI, the SPI is defined as the time from a heel strike of one foot to the next heel strike of the other foot [2]. The sum of two consecutive SPIs is approximately equal to a single SI. Generally, the durations of SI and SPI under self-preferred paces are around 1.0 s and 0.5 s, respectively. Both vary with age and diseases. Extended time series of SI and SPI can be derived using accelerometers [2].

Numerous entropy-based measures of disorder and complexity were proposed and applied to gait analysis. However, some inconsistencies or limitations in finding the relation between the complexity or disorder measurement and gait stability were still recognized and are worth further investigation [2,3].

For example, sample entropy (SE) is a widely used measure of disorder in terms of irregularity [4], which achieves maximum in a completely random series such as white noise. However, SE has been reported sensitive to its input parameters. In addition, the time series for SE analysis has to be long enough to measure the conditional entropy within it, resulting in a long computation time. Thus, to improve the limitations, several new entropy methods such as fuzzy entropy (FE) [5,6], distribution entropy (DistE) [7], dispersion entropy (DE) [8], fluctuation-based DE (FDE) [9], etc., were proposed. In the studies of gait in children [10], the values obtained by using DE and FDE for elderly children were all greater than those for young children [9]. Whereas, the corresponding FE and DistE values consistently decreased from the younger to the elderly children [11,12].

On the other hand, Costa et al. proposed a multiscale entropy (MSE) analysis to measure the complexity of a physiologic time series by measuring the SE value of the time series on a multiple-time-scale basis. Subsequently, many multiscale-based entropy methods for complexity were developed [13,14,15]. According to the MSE, it is the complexity rather than the irregularity that reflects the degree of the healthiness of a biological system through its output physiologic signals. For example, a pathological atrial fibrillation heart rate time series is highly irregular, which results in a low MSE value and a high SE value at a large time scale [16]. This reflects the difference between complexity and irregularity. Since the MSE analysis is a multiscale-based SE, the target time series has to be long enough too for a reliable measure of complexity using the multiscale-based entropies such as the MSE. However, the numbers of strides of the diseased subjects are often less than 300 for safety [14], and the multiscale-based methods are impractical in gait analysis.

However, entropy of entropy (EoE) [17] and average entropy (AE) [18] are other measures of complexity and disorder, respectively. Different from the above measures developed from SE, both EoE and AE are developed from Shannon entropy on the multiscale basis. It has been long considered that the complexity is relatively high for a complex system intermediate between extreme order and disorder [16,19,20,21,22,23]. Accordingly, the expected inverted U curves of complexity (EoE) versus disorder (AE) have been realized for simulated colored noise and real physiological signals such as heart rate signals and static standing postural stability [17,18,21,24]. In addition, both EoE and AE remain effective for short time series.

In the application of the AE to static standing postural stability, the AE values of the center-of-pressure trajectory time series of each of the individuals in testing was found to increase from stable to unstable balance conditions [24]. The study suggested that disorder measurement is useful in a nonautonomically controlled nervous system “such as the static standing, which is under one’s own consciousness”, and complexity measurement is useful in an autonomic system “such as heart beating” [24]. Since walking is controlled by the nonautonomic nervous systems (the somatic nervous system), disorder measurement is also expected useful in gait analysis.

In this study on walking stability, we apply the EoE and the AE to three gait databases to investigate whether the disorder is better than the complexity to measure gait stability. The stable versus unstable conditions in the three gait databases are as follows: (1) healthy subjects walking with eyes open versus closed; (2) healthy subjects versus patients with Parkinson’s disease (PD); (3) healthy subjects versus patients with lateral sclerosis (ALS), Huntington’s disease (HT), and PD. For comparison, the AE values of all the SI and the SPI time series from the three databases are to be compared with those measured by using the SE, the FE, the DE, the FDE, and the DistE. The corresponding means and the standard deviations (SD) of the time series were presented to show the difference between the traditional metrics and the entropy-based method in gait analysis.

## 2. Materials

Three databases of 10 SPI and 79 SI time series were used for the study of walking stability. These databases also provide the acceleration data of individuals walking continuously on an obstacle-free path, from which the SPI and the SI time series were derived.

The SPI time series of the first database (D1) were collected from 10 healthy participants (36.1 ± 18.3 years old). Specifically, each participant walked three laps of a 400-meter-long athletic field with his/her eyes open and closed, respectively. All participants walked at their preferred gait velocities from the same starting point. They took enough rest between their walks with eyes open and closed, individually. When they walked with their eyes closed, every one of them was accompanied by a laboratory staff to prevent from falling or deviating from the track by verbal instructions. An accelerometer (ADXL335 3-axis Digital Gravity Sensor, Analog Devices, Norwood, MA, USA) was secured tightly at the sacrum of each of the participants by using a bandage. The acceleration measurement range of the accelerometer is ±3 g with a resolution of 3 mg, where g is the abbreviation of the acceleration of gravity. The acceleration data were sampled at 100 Hz and saved in a memory card on a programmed Arduino UNO board. The SPI time series were derived from the acceleration data for analysis [25,26,27]. As a result, the 10 healthy participants walked for 1600–2200 steps with intervals around 0.5 s, individually. The methodology of this study was approved by the Human Ethics Research Board at National Chiao Tung University in Taiwan.

The SI time series of the other two databases were downloaded from the online databases on PhysioNet [28]. One is the Gait in Aging and Disease Database on PhysioNet, which includes the data regarding gait variations with age and diseases. This database, called D2 in this study, contained the SI time series of 10 healthy individuals and 5 senior patients with PD. Specifically, the healthy group consisted of 5 young adults and 5 seniors. The healthy subjects walked in a roughly circular path for 15 min, resulting in 700–800 strides with intervals around 1.0 s. The patients walked in a long hallway for 6 min, resulting in 200–280 strides with intervals of 1.0–1.2 s. The other is the Gait Dynamics in Neuro-Degenerative Disease Database on PhysioNet [29]. This database, called D3 in this study, contained the SI time series of 16 healthy individuals and 48 patients with neuro-degenerative disease. Specifically, the pathologic group was composed of 20, 15, and 13 patients with HT, PD, and ALS, respectively. The healthy subjects and patients walked for 200–270 strides with intervals around 1.0 s and 120–280 strides with intervals of 1.0–1.5 s, respectively. Additional details on these databases are provided in [28,29].

## 3. Methods

### 3.1. Entropy of Entropy (EoE) and Average Entropy (AE) Analyses

For analyzing a time series {*x_i_*} = {*x*_1_, …, *x_N_*} of length N, the algorithms of the AE and EoE methods consist of three steps. The first two steps are the same for both entropies. The first step is to divide a time series for analysis into consecutive and non-overlapping windows with an equal length of τ, referred to as a scale factor. Each window is in the form of *w_j_*^(*τ*)^ = {*x*_(*j*−1)*τ*+1_, …, *x*_(*j*−1)*τ*+*τ*_}, where *j* is the window index ranging from 1 to *N*/*τ*.

The second step is to calculate the Shannon entropy value of the data within each of the windows *w_j_*^(*τ*)^ as follows. All the 79 SI time series in databases D2 and D3 were examined to obtain the maximal (*T_max_*) and the minimal (*T_min_*) values among them. The range between *T_max_* and *T_min_* in amplitude was divided into *s*_1_ slices of equal width such that each slice represents a state. Over each window, the probability to find a data in each of the slices was obtained. The probability *p_jk_* for a certain data point *x_i_* over window *w_j_*^(^*^τ^*^)^ to occur in slice *k* is thus obtained in the form of
(1)$$p_{jk}=\frac{\mathrm{total}\;\mathrm{number}\;\mathrm{of}\;x_{i}\;\mathrm{over}\;w_{j}^{\left(\tau \right)}\;\mathrm{in}\;\mathrm{slice}\;k}{\tau },$$
where *k* is the slice index ranging from 1 to *s*_1_. Accordingly, the Shannon entropy value of the data in each window was derived from the distribution of the probabilities over the slices. The Shannon entropy value *y_j_*^(*τ*)^ of each window *w_j_*^(*τ*)^ is given by
(2)$$y_{j}{}^{\left(\tau \right)}=-\sum_{k=1}^{S_{1}}p_{jk}(\ln p_{jk}).$$

As a consequence, a new sequence {*y_j_*^{*τ*}^} of length *N*/*τ* consisting of the Shannon entropy values derived from the windows *w_j_*^(*τ*)^ was formed for each of the original SI time series. Similarly, the same procedure was repeated for all the 10 SPI time series in database D1.

Third, the AE value of {*x_i_*} is defined as the average of the Shannon entropy sequence {*y_j_*^(*τ*)^} in the form of
(3)$$\mathrm{AE}(\tau )=\frac{\sum_{j=1}^{N/\tau }y_{j}{}^{\left(\tau \right)}}{N/\tau }.$$

However, the EoE value of {*x_i_*} is defined as the Shannon entropy value of the new sequence {*y_j_*^{*τ*}^}. All the 79 new sequences derived from the 79 SI time series were examined to obtain the maximal (*SE_max_*) and the minimal (*SE_min_*) values among them. The range between *SE_max_* and *SE_min_* was divided into *s*_2_ slices of equal width. The probability of finding data in each of the slices over the new sequence was obtained. The probability *p_l_* for a certain *y_j_*^{*τ*}^ over the sequence {*y_j_*^(*τ*)^} to occur in level *l* is obtained in the form of
(4)$$p_{l}=\frac{\mathrm{total}\;\mathrm{number}\;\mathrm{of}\;y_{j}{}^{\left(\tau \right)}\;\mathrm{over}\;\{y_{j}{}^{\left(\tau \right)}\}\;\mathrm{in}\;\mathrm{level}\;l}{N/\tau },$$
where *l* is the level index ranging from 1 to *s*_2_. Thus, the EoE value of {*x_i_*} is defined as the Shannon entropy value of the Shannon entropy sequence {*y_j_*^(*τ*)^} in the form of
(5)$$\mathrm{EoE}(\tau )=-\sum_{l=1}^{S_{2}}p_{l}(\ln p_{l}).$$

Note that the values of *s*_1_ and *s*_2_ were set at the maximal accuracy in differentiating the healthy from the diseased for databases D2 and D3. Similarly, the same procedure was repeated for all the 10 SPI time series in database D1. For additional details, please refer to [17,18].

Before proceeding the gait analysis by using the AE and the EoE methods, the SI and the SPI time series were filtered by eliminating the points of extreme deviation. Then, the values of the parameters *T_min_*, *T_max_*, *SE_min_*, *SE_max_*, *τ*, *s*_1_, and *s*_2_ were determined according to the procedures as described above. As a result, for the 79 SI time series, *T_min_*, *T_max_*, *SE_min_*, *SE_max_*, *τ*, *s*_1_, and *s*_2_ were found to be 0.5 s, 2 s, 0, 3, 10, 50, and 15, respectively. Similarly, for the 10 SPI time series, *T_min_*, *T_max_*, *SE_min_*, *SE_max_*, *τ*, *s*_1_, and *s*_2_ were found to be 0.25 s, 1 s, 0, 3, 10, 50, and 15, respectively. It is worth noting that the values of *T_min_* and *T_max_* among the SPI time series are half of those among the SI time series since the sum of two consecutive SPIs is equal to a single SI. In addition, the resulting *s*_1_ = 50 in the gait analysis is reasonable in comparison with *s*_1_ = 55 in the heartbeat analysis [17].

At last, the AE and the EoE methods were used to measure the degrees of disorder and complexity of each of the SPI and the SI time series of the three databases. For further algorithm details and MATLAB codes, please refer to [17,18,30].

### 3.2. ESample Entropy (SE), Fuzzy Entropy (FE), Dispersion Entropy (DE), Fluctuation-Based Dispersion Entropy (FDE), and Distribution Entropy (DistE) Analyses

In this study, the state-of-the-art entropies including SE, FE, DE, FDE, and DistE were utilized for further comparison. All these entropies measure the irregularity, namely, the disorder of data. Before proceeding by using these entropy methods, the SI and the SPI time series were filtered by eliminating the points of extreme deviation. Two sets of input parameters were used to derive the entropy values of the same time series in D1, D2, and D3 for comparison. According to the commonly used default values suggested in the original papers [30], the first set of input parameters (P1D) were set as follows: (1) SE: The embedding dimension and the similarity threshold were 2 and 0.15, respectively; (2) FE: The embedding dimension was 2, and the step and the width of the fuzzy exponential function were 2 and 0.3, respectively; (3) DE and FDE: The mapping method, embedding dimension, the number of classes, and the time delay were normal cumulative distribution functions, 3, 6, and 1, respectively; (4) DistE: The embedding dimension, the bin number, and the time delay were 2, 512, and 1, respectively.

The second set of input parameters (P2I) were tuned to improve the capability in differentiating the healthy from the pathologic groups. As the number of classes in FDE was set equal to the bin number in DistE, 512, the overall performance under the new set of input parameters P2I is better than that under P1D, which is commonly used. The differences between the P2I and the P1D were listed as follow: (1) SE: The similarity threshold was changed to 0.0015; (2) FE: The width of the fuzzy exponential function was changed to 0.003; (3) DE: The number of classes was changed to 128; (4) FDE: The number of classes was changed to 512. For additional details about the algorithms, please refer to [5,6,7,8,9,12,30,31].

### 3.3. The Performance Indices

To quantify the performances of these entropy methods, we make use of the following performance indices accuracy *Acc_D_*_1_, *Recall*, *Precision*, and *F* score according to the results obtained by applying the traditional metrices of mean and SD, as well as the entropy analyses of AE, SE, FE, DE, FDE, and DistE to D1, D2 and D3, respectively. For database D1, the accuracy was obtained in the form
(6)$$Acc_{D1}=\frac{n_{trend}}{N_{d}},$$
where *N_d_* is the total number of subjects of the database and *n_trend_* is the number of the participants who exhibit a relatively high entropy value when walking with eyes closed and a relatively low entropy value when walking with eyes open, individually.

For databases D2 and D3, quadratic discriminant analysis (QDA) was applied to classify the entropy values of the SI time series of the healthy and pathologic groups [32]. In the training phase of each entropy method, we calculated the mean and the standard deviation of all the entropy values associated with the healthy and pathologic groups. Thus, a Gaussian distribution was fitted to the distribution of entropy values in each group. Moreover, two different Gaussian curves, corresponding to the two groups, were obtained. As a consequence, the intersection point of the two Gaussian curves determined the threshold for the classification of the entropy values into the two groups. In the testing phase of each entropy method, we adopted the leave-one-out cross validation (LOOCV) of multi-class classification problems [33]. Three indices under LOOCV were given by
(7)$$Recall=\frac{TP}{TP\;+\;FN},$$
(8)$$Precision=\frac{TP}{TP+FP},$$
(9)$$F=\frac{2\cdot Recall\cdot Precision}{Recall+Precision},$$
where *TP*, *FP*, and *FN* mean true positive, false positive, and false negative, respectively.

Besides, the values of the performance indices for the various groups in D2 and D3 were calculated for comparison based on the sample size guideline of equal group probability for two-group discriminant analysis [34]. Specifically, in D2, the results of the 5 seniors (SN) from the healthy group were compared with those of the 5 senior patients of PD. In D3, the results of all 16 healthy subjects (H) were compared with those of the 20 patients of HT, the 15 patients of PD, and the 13 patients of ALS, respectively.

## 4. Results

Figure 1a,b illustrate the AE and the EoE values of the 10 SPI time series of database D1 at *τ* = 10, respectively. Each of the 10 participants exhibited a relatively high AE value, labeled in red, when walking with eyes closed and a relatively low AE value in green when walking with eyes open, respectively. The trend of the change due to the visual feedback between eyes closed and open is completely consistent on the AE values for the 10 participants, individually. However, the trend is not completely consistent on the EoE values since 2 of the 10 participants exhibited an opposite change, as shown in Figure 1b. The experimental result indicates that the better the control in walking through visual feedback, the higher the walking stability and the smaller the resulting AE.

Figure 2a,b illustrate the AE and the EoE values of the 15 SI time series of database D2 at *τ* = 10, respectively. The 15 subjects were composed of 10 healthy individuals and 5 patients with PD. In Figure 2a, the healthy group presents relatively low AE values, labeled in green, while the pathologic group presents relatively high AE values in red, in general. However, the similar trend does not exist in the distribution of the EoE values, as shown in Figure 2b. The result suggests that the AE rather than the EoE is the appropriate measure to differentiate the healthy from the diseased with pathological abnormalities.

Similar to Figure 2, Figure 3a,b also illustrate the AE and the EoE values of the 64 SI time series of database D3 at *τ* = 10, respectively. The 64 subjects consisted of 16 healthy individuals and 48 patients with ALS, HD, and PD. In Figure 3a, the healthy group presents relatively low AE values, labeled in green, while the pathologic group presents relatively high AE values in red, in general. However, the similar trend does not exist in the distribution of the EoE values, as shown in Figure 3b. The results also suggest that the AE is the suitable measure to differentiate the healthy from the diseased with pathological abnormalities.

Figure 4 depicts the plot of the EoE versus the AE values of the 79 SI time series of databases D2 and D3. The comparison of the EoE and AE plot exhibited an inverted U curve [21], which is another example of a complexity versus disorder inverted U curve for physiologic signals [17,18,24,35], as expected. We find that a threshold of AE_th_ = 1.06, the dashed line in the figure, is optimal to differentiate the healthy from the diseased. Note that only 1 out of the 53 high AE values in red is below the threshold, with 20 out of the 26 low AE values in green. Thus, the overall accuracy of 91.1% (=72/79) is maximal in differentiating the healthy from the diseased.

Applying the same threshold of AE = 1.06 as the criterion of stability into Figure 3a, we find that only 5 out of the 16 low AE values in green is above the threshold with 47 out of the 48 high AE values in red. Thus, the accuracy is 90.6% (=58/64) for database D3. Similarly, the same threshold will result in an accuracy of 93.3% (=14/15) for database D2 in Figure 2a.

Table 1 lists the resulting values of the performance indices *Acc_D_*_1_, *Recall*, *Precision*, and *F* obtained by applying the traditional metrices of mean and SD, as well as the analyses of AE, SE, FE, DE, FDE, and DistE to D1, D2, and D3, respectively. As expected, the overall performances of the 5 disorder entropies under the new set of input parameters P2I are better than those under P1D, which is commonly used. By comparison, the AE and the DistE exhibit the best and the second-best performances for all D1, D2, and D3, respectively. Yet, the overall performances of the mean and the SE are relatively poor.

## 5. Discussion

Second only to the AE analysis, the DistE exhibited the best performance under P1D among the 5 disorder measurements of SE, FE, DE, FDE, and DistE; the DistE exhibits the best performance under P1D. As we tuned the number of classes in FDE equal to the bin number in DistE, 512, the overall performances of the 5 disorder measurements under the new set of input parameters P2I are significantly improved in comparison with those under P1D, which is commonly used. Besides, it is worth noting the relative change of the DE and FDE values for the healthy and the pathologic groups under P1D and P2I. This finding is consistent with the results in the previous studies of gait in children. The values obtained by using FE and DistE decreased from the younger to the older children [11,12], while the corresponding DE and FDE values for the older children were all greater than those for the younger children [9]. This is an interesting topic worth further study.

The two traditional metrices of SD and mean are not as good as the AE in differentiating the healthy from the diseased groups in D1, D2, and D3. The main difference between them is the lack of the information on the sequential order of a given time series in the traditional metrices. Recall that the AE method applies the multiscale technique to extract the randomness of a series [18], while the values of SD and mean remains the same after the series is shuffled. Thus, the two traditional metrices are not useful in gain stability analysis.

An inverted U relation between the complexity and the disorder measurements of a series is a characteristic of a physiological time series. The feature was first conceptually proposed by Huberman and Hogg in 1986 [21]. In 2019, we first demonstrated that the plot of complexity measured by EoE versus the disorder measured by AE of the beat-to-beat intervals time series of electrocardiography exhibited a distinct inverted U relation [17]. In 2020, we illustrated that the EoE and the AE values of the center-of-pressure trajectories of the subjects under different kinds of balance conditions exhibited an inverted U relation [24]. In this study, we further showed that the EoE versus AE plot of the 79 SI time series in gait also exhibited an inverted U curve. The result indicates that the AE and the EoE are good measures of disorder and complexity in gait analysis, respectively.

## 6. Conclusions

The AE, a measure of disorder, has been demonstrated to be a good measure of static standing postural stability [13]. This study further investigates the validity of the AE to be a good measure of walking stability in terms of gait stability.

The results of the 10 SPI and the 79 SI time series in gait showed that every one of the 10 participants exhibited a relatively high AE value of the SPI time series when walking with eyes closed while a relatively low AE value when walking with eyes open. The trend is completely consistent on the AE rather than the EoE values. As to the 26 healthy and the 53 diseased subjects, most of the AE values of the SI time series of the diseased subjects were higher than those of the healthy subjects. For comparison, other disorder measures such as mean, SD, SE, FE, DE, FDE, and DistE also show good performances in terms of the accuracy *Acc_D_*_1_, the *Recall*, the *Precision*, and the *F*. Among them, AE, DE, and DistE exhibited better performances than SE, FE, and FDE. On the contrary, the EoE, a complexity measure, could not measure gait stability.

Therefore, the results support that disorder measurement is also valid in gait stability as in static standing postural stability because of the nonautonomically controlled nervous system. Furthermore, the EoE versus AE plot of the 79 SI time series in gait also exhibits an inverted U curve. The shape is another example of the complexity versus disorder relation hypothesized for physiologic signals.

## Figures and Tables

**Figure 1 entropy-23-00412-f001:**
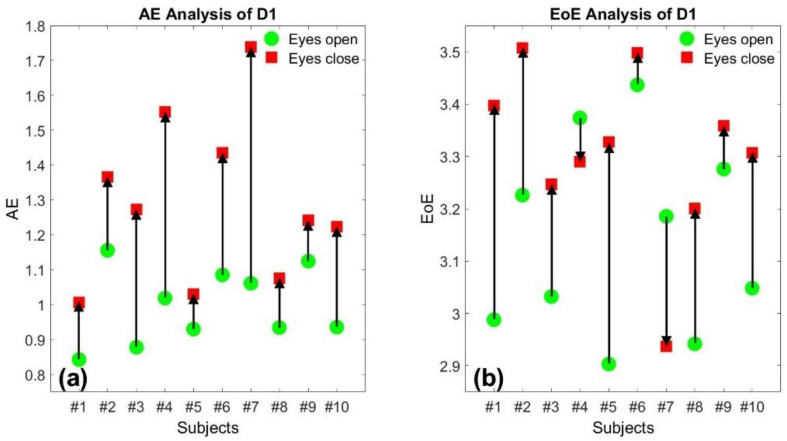
(**a**) The average entropy (AE) and (**b**) the entropy of entropy (EoE) values of the 10-step interval (SPI) time series of the first database (D1) at *τ* = 10, respectively. Each of the 10 participants exhibited a relatively high AE value, labeled in red, when walking with eyes closed and a relatively low AE value in green when walking with eyes open, respectively. The trend of the change due to visual feedback is not completely consistent on the EoE values.

**Figure 2 entropy-23-00412-f002:**
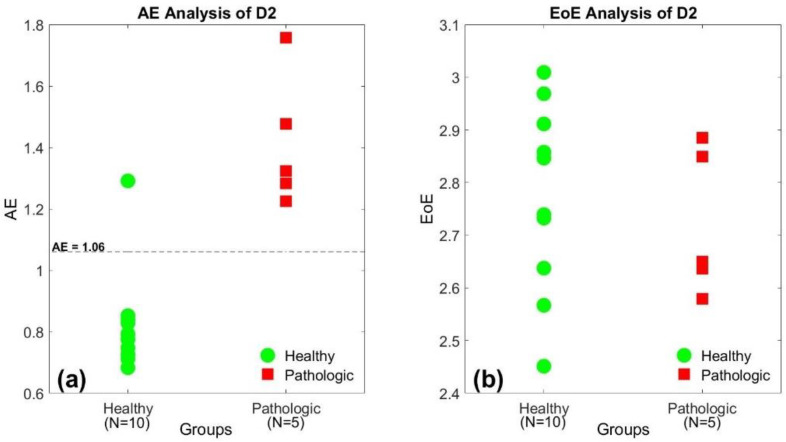
(**a**) The AE and (**b**) the EoE values of the 15-stride interval (SI) time series of the second database (D2) at *τ* = 10, respectively. The healthy group presents relatively low AE values, labeled in green, while the pathologic group presents relatively high AE values in red. The trend does not exist in the distribution of the EoE values.

**Figure 3 entropy-23-00412-f003:**
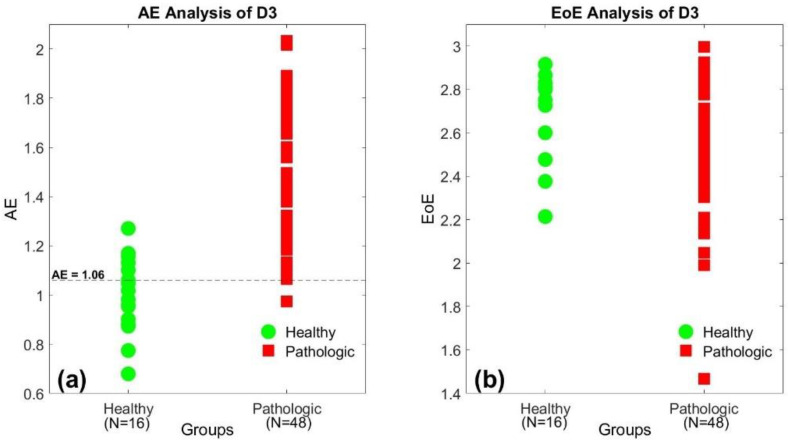
(**a**) The AE and (**b**) the EoE values of the 64 SI time series of the third database (D3) at *τ* = 10, respectively. The healthy group presents relatively low AE values, labeled in green, while the pathologic group presents relatively high AE values in red. The trend does not exist in the distribution of the EoE values.

**Figure 4 entropy-23-00412-f004:**
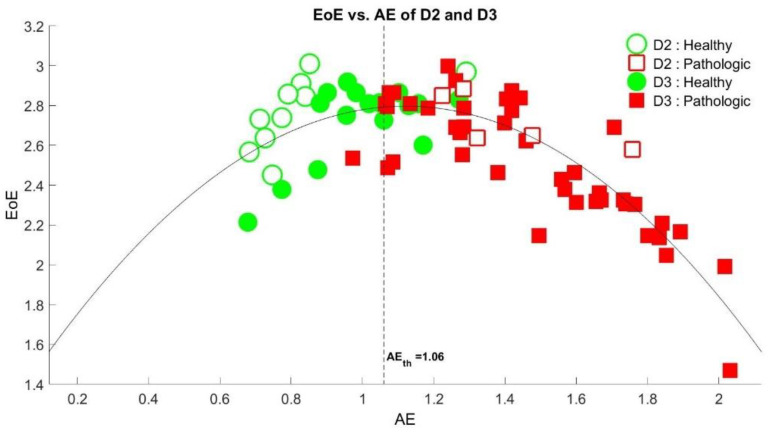
The plot of the EoE versus the AE values of the 79 SI time series of databases D2 and D3 at *τ* = 10, which exhibited an inverted U relation. A threshold of AE_th_ = 1.06, the dash line in the figure, is optimal to differentiate the healthy from the diseased with a maximal overall accuracy of 91.1% (=72/79).

**Table 1 entropy-23-00412-t001:** The performance indices *Acc_D_*_1_, *Recall*, *Precision*, and *F* obtained by applying the traditional metrices of mean and SD, as well as the analyses of AE, SE, FE, DE, FDE, and DistE to D1, D2, and D3, respectively. P1D: the first set of input parameters determined by the commonly used default values suggested in the original papers; P2I: the second set of input parameters for optimal capability in differentiating the healthy from the pathologic groups.

Performance	Mean	SD	AE	DistE	SE P1D/P2I	FE P1D/P2I	DE P1D/P2I	FDE P1D/P2I
*Acc_D_* _1_	50%	80%	100%	80%	70%/90%	50%/60%	60%/80%	50%/60%
*Recall_D_* _2*(SN* vs. *PD)*_	0.6	0.6	1	0.4	0.4/0.6	0.8/0.6	0.8 ^R^/1 ^R^	0.6/1 ^R^
*Precision_D_* _2*(SN* vs. *PD)*_	0.75	1	0.83	0.5	0.33/1	0.67/0.6	0.8 ^R^/1 ^R^	0.6/1 ^R^
*F_D_* _2*(SN* vs. *PD)*_	0.67	0.75	0.91	0.44	0.36/0.75	0.73/0.6	0.8 ^R^/1 ^R^	0.6/1 ^R^
*Recall_D_* _3*(H* vs. *HT)*_	0.65	0.75	0.8	0.8	0.7/0.7	0.75/0.65	0.2 ^R^/0.6 ^R^	0.75/0.65 ^R^
*Precision_D_* _3*(H* vs. *HT)*_	0.87	0.94	0.94	0.89	0.67/0.74	0.68/0.62	1 ^R^/0.92 ^R^	0.63/0.93 ^R^
*F_D_* _3*(H* vs. *HT)*_	0.74	0.83	0.86	0.84	0.68/0.72	0.71/0.63	0.33 ^R^/0.73 ^R^	0.68/0.76 ^R^
*Recall_D_* _3*(H* vs. *PD)*_	0.13	0.47	0.87	0.67	0.53/0.53	0.53/0.6	0.53/0.33 ^R^	0.6/0.33 ^R^
*Precision_D_* _3*(H* vs. *PD)*_	0.29	0.88	0.87	0.83	0.57/0.62	0.8/0.6	0.62/0.45 ^R^	0.82/0.56 ^R^
*F_D_* _3*(H* vs. *PD)*_	0.18	0.61	0.87	0.74	0.55/0.57	0.64/0.6	0.57/0.38 ^R^	0.69/0.42 ^R^
*Recall_D_* _3*(H* vs. *ALS)*_	0.23	0.46	0.69	0.62	0.08/0	0.31/0.38	0.31 ^R^/0.38 ^R^	0/0.38 ^R^
*Precision_D_* _3*(H* vs. *ALS)*_	0.75	0.86	0.9	0.8	0.17/0	0.4/0.83	0.36 ^R^/0.63 ^R^	0/0.63 ^R^
*F_D_* _3*(H* vs. *ALS)*_	0.35	0.6	0.78	0.7	0.11/NaN ^ab^	0.35/0.53	0.33 ^R^/0.48 ^R^	NaN ^ab^/0.48 ^R^

^R^ The relationship between the two distributions of the healthy group and the pathologic group was reversed. ^ab^ The numerical abnormality.

## Data Availability

Not applicable.

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
