# Peer review of "Gait Stability Measurement by Using Average Entropy"

_entropy, 2021, doi:10.3390/e23040412_

Round 1

Reviewer 1 Report

Overall, the writing needs to be improved.

Introduction

  1. “Stride interval” can be stride length or stride time, please be clear about it.
  2. What does “nonautonomic” mean? There are both automatic and non-automatic components in the regulation of walking.
  3. What is the full name of EoE?

Methods

  1. How the correction was provided in eyes-closed walking? What is the trajectory of it? The same as eyes-open walking?
  2. Any issue from fatigue?
  3. Using formula to describe AE and EoE.
  4. What is the length of the SI and SPI series? Please report.
  5.  

Results

  1. What does the colors represent on Figures? Please clarify it on Figure to help readers.
  2. There is an extreme high AE value in Healthy cohort, what is the reason?
  3. So, is AE=1.06 a threshold that can be tested in a new dataset? Can it be used to identify different diseases? Did authors test this?
  4. What is the difference between traditional metrics of gait between groups? Like mean or SD of stride time? If these metrics have similar distribution or difference between groups, what is the additional knowledge that the AE or EoE can provide?

Discussion:

The discussion is weak, and several important aspects were not discussed?

  1. Why EoE and AE were so different here? Any potential neurophysiological mechanisms underneath?
  2. What are the effects of the guidance that was given in EC walking on the outcomes?
  3. Highlighting too much of the accuracy based upon this smaller dataset is weak, better to discuss the underlying neurophysiological mechanisms underlying the gait regulation the AE captured.

Author Response

Clarification for Reviewer #1:

Dear Reviewer,

Thank you greatly for your very kind review of our paper and your valuable comments as well as advice. We have carefully thought about your opinions and discussed them further accordingly. We want to especially thank you for helping us organize the storyline and strengthen our assertion. The issues you suggested are clarified point-by-point in the following text. In the draft, the corresponding modifications were yellow labeled for your reference. If there's anything else we can do to further improve our research, please again give us your very kind advice. Your help is greatly appreciated.

Sincerely,

Sien Chi

Reviewer 2 Report

The paper contains a valuable contribution. The subject is within the scope of the journal and the objective of research is well stated. However, some clarifications about the underlying hypothesis / scope are needed.

In the opinion of this Reviewer the manuscript deserves to be published once the Author takes into account the raised issues.

Introduction / Literature review

  1. The research scope is clear as well as the literature review. Anyway, the authors should better highlight the innovative aspects of their work in the manuscript.

What are the advantages / improvements in the proposed approach, which are not covered by the current studies?

  1. For the sake of readability, at the end of Section 1 the authors should describe how the paper is structured.

Material and methods

  1. Please check the producer of the ADXL335, it could be Analog Devices.

  1. Please specify the unity of measurement for xxx

Minor

  1. Please specify the unity of measurement
  2. The authors should check that all the used acronyms are explained and not repeated every time
  3. Mainly the English is good and there are only a few typos. However, the paper should be carefully rechecked.

Author Response

Clarification for Reviewer #2:

Dear Reviewer,

Thank you greatly for your very kind review of our paper and your valuable comments as well as advice. We have carefully thought about your opinions and discussed them further accordingly. We want to especially thank you for helping us organize the storyline and strengthen our assertion. The issues you suggested are clarified point-by-point in the following text. In the draft, the corresponding modifications were yellow labeled for your reference. If there's anything else we can do to further improve our research, please again give us your very kind advice. Your help is greatly appreciated.

Sincerely,

Sien Chi

Round 2

Reviewer 1 Report

Thanks for the revisions authors made, which addressed most of my concerns. Before publication, I strongly suggest to include the results of traditional metrics of gait in the Results section and add the discussion on the comparison between this kind of metrics and the complexity metrics. This will make the reader from the clinical background understand more of the advantages in using the complexity metrics. 

Author Response

Clarification for Reviewer #1:

Dear Reviewer,

Thank you greatly for your valuable advice. We want to thank you for helping us strengthen our assertion significantly. In the draft, the corresponding modifications were yellow labeled for your reference. If there's anything else we can do to improve our research further, please again give us your very kind advice. Your help is greatly appreciated.

Sincerely,

Sien Chi

* Comment 1:
Thanks for the revisions authors made, which addressed most of my concerns. Before publication, I strongly suggest to include the results of traditional metrics of gait in the Results section and add the discussion on the comparison between this kind of metrics and the complexity metrics. This will make the reader from the clinical background understand more of the advantages in using the complexity metrics. 

Reply 1:
Thanks for the advice. We enriched the Results section by including the results derived from traditional metrics into Table 1. Besides, for the clinical background readers, we modified our comparison method to compare between groups in the database. We thought this modification would help the readers understand more. We also discussed furtherly in the comparison in the Discussion section.
